# SPARSE ATTENTION WITH LEARNING-TO-HASH

**Zhiqing Sun, Yiming Yang**
Language Technologies Institute, Carnegie Mellon University
{zhiqings, yiming}@cs.cmu.edu

**Shinjae Yoo**
Brookhaven National Laboratory
sjyoo@bnl.gov

## ABSTRACT

Transformer has become ubiquitous in sequence modeling tasks. As a key component of Transformer, self-attention does not scale to long sequences due to its quadratic time and space complexity with respect to the sequence length. To tackle this problem, recent work developed dynamic attention sparsification techniques based on Approximate Nearest Neighbor (ANN) methods, where similar queries and keys are allocated to the same hash bucket with high probability. However, the effectiveness of those ANN methods relies on the assumption that queries and keys should lie in the same space, which is not well justified. Besides, some of the ANN methods such as Locality-Sensitive Hashing (LSH) are randomized and cannot fully utilize the available real data distributions. To overcome these issues, this paper proposes a new strategy for sparse attention, namely LHA (Learning-to-Hash Attention), which directly learns separate parameterized hash functions for queries and keys, respectively. Another advantage of LHA is that it does not impose extra constraints for queries and keys, which makes it applicable to the wide range of pre-trained Transformer models. Our experiments on evaluation of the WikiText-103 dataset for language modeling, the GLUE benchmark for natural language understanding, and the Lang-Range-Arena benchmark for multiple tasks (text/image classification, retrieval, etc.) show the superior performance of LHA over other strong Transformer variants.

## 1 INTRODUCTION

The Transformer architecture (Vaswani et al., 2017) has been successfully applied to various tasks, including natural language processing (Vaswani et al., 2017; Devlin et al., 2018; Dai et al., 2019; Liu et al., 2019; Yang et al., 2019), computer vision (Carion et al., 2020; Dosovitskiy et al., 2020), and time series forecasting (Zhou et al., 2020; Wu et al., 2021). Such a success is mainly due to the self-attention component, which enables each token to directly interact with any other tokens in the entire sequence. But self-attention has a quadratic time and space complexity with respect to the sequence length and hence does not scale efficiently to long sequences as a result.

To address this inefficiency problem, one of the solutions is to approximate the full attention matrix with a sparse one, as the $\mathrm{softmax}$ operation is dominated by the largest elements. Some recent efforts focus on dynamic learning of sparse attention patterns via Approximate Nearest Neighbors (ANN) approaches, including Locality Sensitive Hashing (LSH) (Kitaev et al., 2020; Daras et al., 2020) and mini-batch spherical k-means (Roy et al., 2021; Wang et al., 2020a). The queries and keys are hashed or clustered into different buckets, hoping that the queries and keys in the same bucket are similar with a high probability. The effectiveness of those ANN approaches rely on the assumption that the (transformed) query and key vectors should lie in the same space, which could be sub-optimal in dealing with different sparsity patterns, as analyzed in this paper (Section 3.2). Besides, the hash functions in LSH are randomized and data-agnostic, which cannot fully utilize the rich information in real-world data distributions.

In this paper, we address the above limitations of existing ANN-based methods for attention sparsification. Firstly, we analyze two imbalance issues in LSH-produced sparse attention patterns, i.e., unbalanced hash bucket sizes and unbalanced query-key ratios. Secondly, we design a new metric called attention utility to quantify how well the sparse patterns approximate the full attention, and we show that ANN-derived sparse patterns are substantially inferior to their counterparts. Thirdly, we propose a novel solution, namely Learning-to-Hash Attention (LHA), for dynamic attention

sparsification with enhanced model expressiveness. LHA directly optimizes our newly defined attention utility metric in an end-to-end manner via separate learnable hash functions for queries and keys, respectively. As for reducing the computational complexity in the training phase, LHA uses unbiased kernelized attention techniques (Choromanski et al., 2020; Peng et al., 2021) to efficiently approximate the attention utilities. Similar to other sparse attention models (Kitaev et al., 2020; Roy et al., 2021), LHA reduces the overall complexity of self-attention from $\mathcal{O}(N^2)$ to $\mathcal{O}(N^{1.5})$ for sequence length $N$. Our experiments in a wide range of tasks on the evaluation benchmarks for language modeling, natural language understanding, and Long-Range-Arena show that LHA achieves better performance compared to strong transformer baselines.

## 2 RELATED WORK

Related work can be roughly divided into three categories, i.e., location-based sparse attention, content-based sparse attention, and dense approximation of attention, as outlined below.

Location-based sparse attention methods aim to improve the computational efficiency by using pre-specified global or local sparsification patterns over token locations. Liu et al. (2018) proposed to alternate coarse attention layers and local attention layers. Child et al. (2019) used a strided sparse attention pattern in image generation. Sukhbaatar et al. (2019) imposed sparsity based on the predicted temporal window size for each token. Other methods (Zhang et al., 2021; Beltagy et al., 2020; Ainslie et al., 2020; Zaheer et al., 2020) used a pre-specified subset of locations in the input as the global memory, and only allow non-local attentions from this subset to all the other tokens. Location-based sparse attention cannot leverage more flexible content-based interactions among arbitrary positions, as a limitation.

Content-based sparse attention allows more flexible sparse patterns than location-based ones. Malaviya et al. (2018) used sparsemax to obtain a sparse attention matrix, while (Correia et al., 2019) used entmax. These methods require to compute the full attention matrix before sparsification and hence cannot reduce the quadratic computation complexity. Approximate Nearest Neighbor (ANN) methods address this limitation by calculating the content-based sparse patterns in advance (Roy et al., 2021; Kitaev et al., 2020; Daras et al., 2020; Wang et al., 2020a) (which will be discussed more in Section 3.1). Those methods usually apply an ANN module as the shared hash function to both queries and keys. Vyas et al. (2020) and Zhou et al. (2020) sparsified the attention maps by eliminating redundant queries. Tay et al. (2020b) designed a differentiable sorting algorithm of internal representations to enable efficient quasi-global local attention. Contemporary to our work, SparseFinder (Treviso et al., 2021) learns sparse attention patterns that approximate entmax attention, but its bucketing strategies are still based on ANN approaches. In contrast, LHA directly predicts a bucketing strategy (i.e., learnable hash functions) that maximizes the attention utility.

Another line of research explored low-rank or kernelized dense approximation of attention matrices, instead of computing the attention scores exactly for only a few pairs. Wang et al. (2020b) applied a low-rank decomposition to the attention matrix. Xiong et al. (2021) approximated the attention matrix with Nyström approximation. Katharopoulos et al. (2020) utilized the association property of Key-Query-Value multiplication and reduce the quadratic complexity to linear complexity with kernelized approximation to the softmax operation. Choromanski et al. (2020) and Peng et al. (2021) further proposed an unbiased approximation of softmax with random Fourier features.

Our work in this paper is directly related to the second category, i.e., content-based attention sparsification. Specially, we address the limitations of existing ANN-based methods by modeling queries and keys in separate vector spaces and by proposing a novel approach to learn the hash functions for attention sparsification.

## 3 RE-EXAMINATION OF CONTENT-BASED SPARSE PATTERNS

### 3.1 PRELIMINARY

The self-attention mechanism (Vaswani et al., 2017) can be formulated as the weighted sum of the value vectors $V \in \mathbb{R}^{N \times d_h}$ where the weights are calculated using query vectors $Q \in \mathbb{R}^{N \times d_h}$ and

key vectors $K \in \mathbb{R}^{N \times d_h}$ as:

$$\text{Attention}(Q, K, V) = A \cdot V = \text{softmax}\left(\frac{QK^T}{\sqrt{d_h}}\right) \cdot V, \tag{1}$$

where $A$ denotes the matrix of normalized attention weights, $d_h$ is the dimension of hidden representations, and $N$ is the sequence length. self-attention has a quadratic time and space complexity with respect to the sequence length and hence does not scale efficiently to long sequences.

Content-based sparse attention methods usually apply randomized hash functions or a clustering algorithm to queries $\{Q_i\}$ and keys $\{K_j\}$, and hope that similar queries and keys are hashed or clustered into the same bucket. The queries can thus only attend to the keys if both are in the same bucket. Formally, a content-based sparse attention strategy with $B$ hash buckets is defined as:

$$\text{Sparse-Attention}(Q_i, K, V) = \sum_{j:h_Q(Q_i)=h_K(K_j)} \bar{A}_{ij} V_j, \tag{2}$$

where $h_K, h_Q : \mathbb{R}^{d_h} \mapsto [B]$ are the hash functions for keys and queries, and $\bar{A}_{ij} \propto A_{ij}$ is the re-normalized attention weights such that $\forall i$, $\sum_{j:h_Q(Q_i)=h_K(K_j)} \bar{A}_{ij} = 1$. In general (Kitaev et al., 2020; Roy et al., 2021), calculating the hash function and performing local attention for each query have the time complexity of $\mathcal{O}(B)$ and $O(N/B)$, respectively. Thus, the overall complexity[1] of self-attention can be reduced from $\mathcal{O}(N^2)$ to $\mathcal{O}(N \cdot B + N^2/B) \approx \mathcal{O}(N^{1.5})$ when $B \approx N/B \approx \sqrt{N}$.

Since the hash functions are not differentiable, Approximate Nearest Neighbor (ANN) methods are used to derive an effective content-based hash function. Reformer (Kitaev et al., 2020) applies Locality Sensitive Hashing (LSH) to the tied queries and keys, where several hyper-planes are randomly generated to divide tokens into different buckets. SMYRF (Daras et al., 2020) improves Reformer by introducing asymmetric transformation to queries and keys, i.e.,

$$F(Q_i) = \left[Q_i; 0; \sqrt{M_Q^2 + M_K^2 - ||Q_i||_2^2}\right], \quad G(K_j) = \left[K_j; \sqrt{M_Q^2 + M_K^2 - ||K_j||_2^2}; 0\right], \tag{3}$$

where $M_Q = \max_{Q_i} ||Q_i||_2$ and $M_K = \max_{K_j} ||K_j||_2$, such that $||F(Q_i) - G(K_j)||_2^2 = const - Q_i \cdot K_j$. Routing Transformer (Roy et al., 2021) and Cluster-former (Wang et al., 2020a) use mini-batch spherical k-means to partition tokens into different clusters.

## 3.2 BUCKET IMBALANCE ISSUES

Previous content-based sparse attention models take it for granted that the ANN-derived sparse pattern can effectively approximate the full attention. However, it is only verified via empirical evaluation on the down-stream tasks yet, which cannot reflect the true attention map approximation ability. Notice that there are two necessary conditions for sparse attention to work effectively and efficiently:

1. The number of queries and the number of keys in each bucket should be reasonably balanced, as queries should attend to enough keys to get a good approximation of the full-attention;

2. The bucket sizes should be nearly equal in order to effectively reduce the overall complexity.

We first analyze how badly the two conditions would be violated by LSH[2], a typical ANN method. We apply LSH to 10 attention heads in the $3^{rd}$ layer of a Transformer[3] pre-trained on language modeling and obtain the results shown in Figure 1 (up). We can see that the imbalance issue not only exists in the query-key ratios, but also in the bucket sizes. To go a step further, we apply LSH to all $16 \times 10 = 160$ attention heads in the pre-trained Transformer and find that around **61.3%** buckets have the query-key imbalance problem, where the query-key ratios are either greater than 2:1 or smaller than 1:2. Around **35.9%** buckets have the bucket size imbalance problem, where the bucket

---

[1]Notice that we only consider the setting of single-round hashing in our paper, but our analysis and the proposed LHA method can be generalized to the multi-round hashing setting.

[2]We use the same LSH technique as in (Kitaev et al., 2020), except that we do not impose extra constraints to queries and keys. This is because we would like to develop a plug-and-play replacement for dense attention layers without imposing extra constraints for queries and keys.

[3]The detailed experimental setting can be found in the appendix.

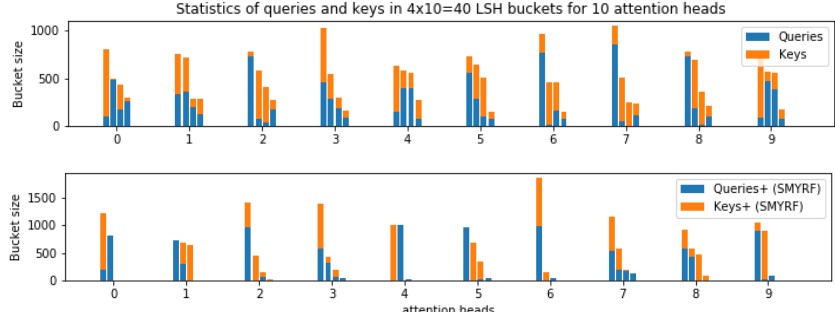

Figure 1: We show the unbalanced bucket sizes and unbalanced query-key ratios with statistics of 10 attention heads in the $3^{rd}$ layer of a pre-trained Transformer. For each head, we assign the queries and keys of first 1024 tokens in the WikiText-103 validation data into 4 LSH buckets. The buckets are sorted for each attention head according to total bucket sizes (i.e., #query + #key).

sizes are twice greater than half smaller than the expected bucket size of $512$. Clearly, neither of the two aforementioned conditions are well satisfied in this LSH-sparsified Transformer model.

There can be several possible reasons that cause the imbalance problem: 1) the Euclidean distance metric in ANN methods does not monotonically decrease with the dot-product metric used in attention mechanism; 2) the queries and keys are from different distribution and not normalized, and thus restricts the effectiveness of ANN methods. To investigate the root cause, we apply the SMYRF asymmetric transformation (Equation 3) to queries and keys, which creates a monotonic relation between Euclidean distances and dot-products. The new analysis results are shown in Figure 1 (down). We can see that the asymmetric transformation would only exacerbate the imbalance problem, with respect to both query-key ratios and bucket sizes. Therefore, we can conclude that the root cause of the imbalance problem is the mismatch between the query and key distributions, that could be further magnified by the asymmetric transformation.

### 3.3 PROPOSED METRIC: ATTENTION UTILITY

The above analysis shows the imbalance problem in ANN-derived hashing strategies. To further quantify the approximation quality of sparse attention patterns, we utilize the concept of Attention Biclustering from (Daras et al., 2020), which considers a practical case (Kitaev et al., 2020; Roy et al., 2021) where all clusters strictly contain the same number of queries and keys:

$$\forall b \in [B], \quad |\{Q_i \mid h_Q(Q_i) = b\}| = |\{K_j \mid h_K(K_j) = b\}| = \frac{N}{B}. \tag{4}$$

We denote with $\mathcal{C}$ the set of all possible assignments in $B$ balanced non-overlapping clusters: $\mathcal{C} = \{\mathcal{C}_1, \mathcal{C}_2, \ldots, \mathcal{C}_T\}$, where $T$ is the number of possible assignments. We can then define the attention utility AU of each assignment $C_t$ as:

$$\text{AU}(\mathcal{C}_t) = \sum_{i,j:(Q_i,K_j)\in\mathcal{C}_t} A_{ij}, \quad \text{where} \quad \mathcal{C}_t = \{(Q_i, K_j)|h_Q^{C_t}(Q_i) = h_K^{C_t}(K_j)\}. \tag{5}$$

The attention utility represents the aggregation of the sparse attention weights in an assignment. As we should keep as much sparse attention weights as possible to better approximate a full attention map, attention utility quantifies how well a sparse attention approximates the full one. In fact, we can show that it is computationally intractable to find the optimal attention utility:

**Theorem 1.** *Finding the assignment that achieves the optimal attention utility, i.e.,*

$$\arg\max_{\mathcal{C}_t \in \mathcal{C}} \text{AU}(\mathcal{C}_t) = \sum_{(Q_i,K_j)\in\mathcal{C}_t} A_{ij} \tag{6}$$

*is NP-hard.*

This theorem is a corollary of the NP-hardness of Attention Biclustering in (Daras et al., 2020), and it motivates us to develop a learning-based approach to optimize the attention utility. Notice that similar

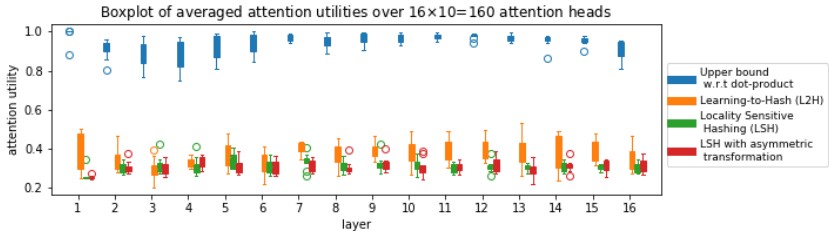

Figure 2: We show the attention utility evaluation of four different sparse attention patterns. For L2H and LSHs, we assign the queries and keys of first 1024 tokens in the WikiText-103 validation data into 4 hash buckets. The causal mask is not used when calculating the attention weights.

to other sparse attention models (Kitaev et al., 2020; Roy et al., 2021), the proposed attention utility metric currently does not take causal masks into consideration. Developing a new metric that can take arbitrary attention masks into account is left for future work.

To measure the attention utility of ANN-derived sparse attention patterns, we follow (Kitaev et al., 2020; Daras et al., 2020) and adaptively set hash boundaries to create balanced clusters. The analyzed models include LSH, LSH with asymmetric transformation (i.e., SMYRF), and our proposed Learning-to-Hash (L2H) method which will be introduced in the next section. We also calculate the empirical upper bound by aggregating the attention from each query to its top keys w.r.t dot-products. We present a box-plot of averaged attention utilities for all 16 Transformer layer in Figure 2. We can see that the LSH methods obtain significantly less attention utilities than the upper bound or L2H. As a conclusion, the ANN-derived sparse patterns are arguably sub-optimal when used as hash functions for sparse attention. We believe this also partially explains why most of these approaches need to impose extra constraints (e.g. queries and keys are tied or normalized). In the next section, we propose an alternative choice of the hash functions to tackle this problem.

## 4 LEARNING-TO-HASH ATTENTION (LHA)

We propose a novel Learning-to-Hash Attention model in this section. The key idea is to learn separate parameterized hash functions for queries and keys, respectively, thus the sparse pattern in LHA can be no longer limited to distance-based hash functions such as LSH or online k-means and adapt to the mismatch of query and key distributions. Besides, we empirically find that LHA can be used as a plug-and-play replacement for dense attention layers, which makes LHA applicable to the wide range of pre-trained Transformer models.

### 4.1 LEARNING-TO-HASH FOR SPARSE ATTENTION

We first remind the readers of our definitions of the sparse attention:

$$\text{Sparse-Attention}(Q_i, K, V) = \sum_{j:h_Q(Q_i)=h_K(K_j)} \bar{A}_{ij} V_j \qquad (7)$$

where $h_K, h_Q : \mathbb{R}^{d_h} \mapsto [B]$ are the hash functions for keys and queries, and $\bar{A}_{ij} \propto A_{ij}$ is the re-normalized attention weights such that $\forall i, \sum_{j:h_Q(Q_i)=h_K(K_j)} \bar{A}_{ij} = 1$. Inspired by the Learning-to-Hash methods (Wang et al., 2017), we implement the learnable hash functions $h_K, h_Q : \mathbb{R}^{d_h} \mapsto [B]$ by defining parameterized functions $H_Q, H_K : \mathbb{R}^{d_h} \mapsto \mathbb{R}^B$, such that:

$$h_Q(Q_i) = \underset{b \in \{1,...,B\}}{\arg\max} \, [H_Q(Q_i)]_b, \quad h_K(K_j) = \underset{b \in \{1,...,B\}}{\arg\max} \, [H_K(K_j)]_b, \qquad (8)$$

where $H_Q$ and $H_K$ can be any parameterized functions such as MLPs.

Notice that our formulation is a generalization of the non-learnable sparse attention mechanisms in several previous content-based sparse attention models. To reproduce the symmetric LSH-based sparse attention scheme in Reformer (Kitaev et al., 2020) or the mini-batch k-means scheme in Routing Transformer (Roy et al., 2021), we can set:

$$H_Q(x) = H_K(x) = x[\mathbf{R}; -\mathbf{R}] \quad \text{or} \quad H_Q(x) = H_K(x) = x\boldsymbol{\mu} \qquad (9)$$

where $\mathbf{R} \in \mathbb{R}^{d_h \times \frac{B}{2}}$ is a random matrix, $[\cdot; \cdot]$ denotes the concatenation of two vectors, and $\boldsymbol{\mu} \in \mathbb{R}^{d_h \times B}$ are the exponentially moving averaged cluster centroids shared by keys and queries. The asymmetric transformation in SMYRF (Daras et al., 2020) can also be reproduced by:

$$H_Q(x) = \mathbb{1}\left(\left\lfloor \frac{F(x) \cdot \mathbf{a} + \mathbf{b}}{r} \right\rfloor\right), \qquad H_K(x) = \mathbb{1}\left(\left\lfloor \frac{G(x) \cdot \mathbf{a} + \mathbf{b}}{r} \right\rfloor\right) \tag{10}$$

where $\mathbf{a}$ is a random vector, b is a random scalar, $r$ is a scalar parameter, $\mathbb{1}(\cdot) : [B] \mapsto \{0, 1\}^B$ is the one-hot operation, and $F(\cdot)$ and $G(\cdot)$ are defined as in Equation 3.

## 4.2 OPTIMIZATION OBJECTIVE FOR LEARNING-TO-HASH

Previous ANN-derived sparse attention models construct their hash functions by either randomization or k-mean clustering. In contrast, our hash functions are fully learnable. However, as we use the $\arg\max$ operation to get the hashing buckets, the parameterized projections in our hash functions cannot be trained in an end-to-end manner. To resolve this issue, we guide the training of our learnable hash functions directly by our proposed attention utility (See Equation 5). Notice that the attention utility of each query $Q_i$ can be written as $\sum_{j:h_Q(Q_i)=h_K(K_j)} A_{ij}$. We first calculate the possible attention utilities $\boldsymbol{\psi}^{(i)}$ that query $Q_i$ can obtain when allocated to all $B$ hash buckets:

$$\boldsymbol{\psi}^{(i)} = \left( \sum_{j:h_K(K_j)=0} A_{ij}, \cdots, \sum_{j:h_K(K_j)=B-1} A_{ij} \right) \tag{11}$$

Ideally, our hash function would allocate $Q_i$ to the bucket that maximize the attention utility, such that $Q_i$ can attend to the most salient keys:

$$h_Q(Q_i) = \arg\max_{b \in [N_b]} \boldsymbol{\psi}_b^{(i)} = \arg\max_{b \in [B]} \sum_{h_K(K_j)=b} A_{ij} \tag{12}$$

To achieve this goal, since $\boldsymbol{\psi}^{(i)}$ is naturally a normalized categorical distribution, we can simply convert $H_Q(Q_i)$ into a distribution and use the KL divergence between the predicted distribution $\mathrm{softmax}(H_Q(Q_i))$ and the desired distribution $\boldsymbol{\psi}^{(i)}$ as the optimization objective:

$$\mathcal{L}_{Q_i} = KL(\boldsymbol{\psi}^{(i)} \| \mathrm{softmax}(H_Q(Q_i))) \tag{13}$$

On the other hand, the possible attention utilities $\boldsymbol{\psi}'^{(j)}$ that a key $K_i$ obtains when allocated to all $B$ buckets can be written as:

$$\boldsymbol{\psi}'^{(j)} = \left( \sum_{i:h_Q(Q_i)=0} A_{ij}, \cdots, \sum_{i:h_Q(Q_j)=B-1} A_{ij} \right) \tag{14}$$

While $\boldsymbol{\psi}'^{(j)}$ is no longer a normalized distribution, we can normalize it and similarly define the optimization objective $\mathcal{L}_{K_j}$ also as a KL divergence.

During the training stage, our final optimization objective is a convex combination of the task-specific objective and the learning-to-hash objectives (i.e., $\{\mathcal{L}_{Q_i}\}$ and $\{\mathcal{L}_{K_j}\}$) for all queries and keys. Figure 3 in the appendix illustrates the joint training diagram of LHA.

## 4.3 APPROXIMATE ATTENTION UTILITIES

The remaining problem is how to efficiently compute $\boldsymbol{\psi}^{(i)}$ and $\boldsymbol{\psi}'^{(j)}$. A naive way requires calculating the dot products between all query-key pairs with $\mathcal{O}(N^2)$ complexity. Inspired by the recent advances in kernelized attention (Katharopoulos et al., 2020; Choromanski et al., 2020; Peng et al., 2021), we use random Fourier features (Rahimi et al., 2007; Choromanski et al., 2020) to approximate $\boldsymbol{\psi}^{(i)}$ in an unbiased manner. Let us define $\boldsymbol{\phi} : \mathbb{R}^{d_h} \mapsto \mathbb{R}^{2D}$ as the Positive Randomized Features (PRFs) (Choromanski et al., 2020) such that

$$\mathbb{E}[\boldsymbol{\phi}(\mathbf{x}) \cdot \boldsymbol{\phi}(\mathbf{y})] = \exp\left(\frac{\mathbf{x}^T \mathbf{y}}{\sqrt{d_h}}\right) \tag{15}$$

We can approximate $\psi^{(i)}$ with the following formula:

$$\psi_b^{(i)} = \sum_{j:h_K(K_j)=b} A_{ij} \propto \sum_{j:h_K(K_j)=b} \exp\left(\frac{Q_i \cdot K_j}{\sqrt{d_h}}\right) \propto \mathbb{E}\left[\phi(Q_i) \cdot \sum_{j:h_K(K_j)=b} \phi(K_j)\right] \quad (16)$$

for each bucket $b \in [B]$. Since $h_K(K_j)$ does not change for the queries, we can simply pre-compute $\sum_{j:h_K(K_j)=b} \phi(K_j)$ once to save computation, which reduces the complexity of computing $\{\psi^{(i)}\}$ for all queries from $\mathcal{O}(N^2)$ to $\mathcal{O}(N \cdot B) \approx \mathcal{O}(N^{1.5})$. The attention utilities for keys $\psi'^{(j)}$ can be efficiently approximated in a similar way[4]:

$$\psi_b'^{(i)} = \sum_{i:h_Q(Q_i)=b} A_{ij} \propto \mathbb{E}\left[\left(\sum_{j:h_Q(Q_i)=b} \frac{\phi(Q_i)}{\phi(Q_i) \cdot \sum_{j=1}^N \phi(K_j)}\right) \cdot \phi(K_j)\right] \quad (17)$$

### 4.4 IMPLEMENTATION DETAILS

Practical content-based sparse attention models require that each hash bucket has the same size, which is crucial in terms of computational efficiency on modern hardwares. Therefore, we follow Roy et al. (2021) and sort the tokens with regard to normalized hash scores, i.e., softmax($H_Q(Q_i)$) and softmax($H_K(K_i)$), in each bucket. The hash bucket membership is then determined by the top-$k$ threshold, where $k = \frac{N}{B}$ is the bucket size. Since such a bucketing strategy no longer guarantees the validation of attention bi-clustering, as each query or each key can be assigned to zero or more than one clusters, we further empirically enlarge the hash bucket sizes by $\sqrt{2}\times$ to increase the recall of queries and keys. A pseudo-code implementation for LHA can be found in the appendix.

In causally masked attentions, the queries cannot attend to keys behind, as is usually the case in language modeling. When we use separate hash functions $\{h_Q(Q_i)\}$ and $\{h_K(K_j)\}$ for queries and keys, respectively, it is possible that for some query $Q_i$, there exists no such key $K_j$ in the same bucket that $j \le i$. This would cause serious numerical instability in our implementation. To tackle this problem, inspired by Kitaev et al. (2020) and Roy et al. (2021), we tie the key hashes with query hashes in the case of causal attentions by constructing a joint hash function:

$$h_K(K_i) = h_Q(Q_i) = \arg\max_{b \in [N_b]} [H_Q(Q_i)_b + H_K(K_i)_b] \quad (18)$$

We find that this strategy empirically works better than using other tricks to fix this "no-attention-target" problem. Our solution is different from Routing Transformer (Roy et al., 2021) or Reformer (Kitaev et al., 2020), which impose an extra constraint that queries and keys are tied.

## 5 EXPERIMENTS

In this section, we conduct experiments to verify the effectiveness of our approach on several benchmark datasets covering language modeling, natural language understanding, and Long-Range-Arena. Due to the space limitations, the detailed hyper-parameter settings are presented in the appendix.

### 5.1 LANGUAGE MODELING

`Wikitext-103` (Merity et al., 2016) is a large-scale dataset for testing long term dependencies in word-level language models. It contains 103M training tokens from 28K articles, with an average length of 3.6K tokens per article, which allows testing the ability of long-term dependency modeling. We use this dataset as a probe dataset to perform various ablations to tease apart the effect of various hyper-parameter choices on the model performance.

We follow the `base` setting of the state-of-the-art Transformer-XL (Dai et al., 2019) model, which contains 16 Transformer layers with 10 heads per layer. For local attentions, we use the relative positional encoding (Dai et al., 2019), while for non-local attentions (i.e., LSH or LHA), we use the

---

[4]Please refer to (Choromanski et al., 2020) for how PRFs stabilize attention renormalization.

Table 1: Ablation studies on the WikiText-103 validation data in the `base` setting. Lower perplexity (PPL) is better. All the models have a total of 16 attention layers and 10 heads. Non-Local (NL, i.e., LSH or LHA) layers when present are always added at the top of the model. Attention size denotes either local window size or hash bucket size. † denotes that the results are taken from Transformer-XL (Dai et al., 2019).

|  | label | NL Heads | NL Layers | Hash Func. | Att. Size | #Param | Valid PPL | Test PPL |
|---|---|---|---|---|---|---|---|---|
| Local Transformer | - | 0 | 0 | - | 768 | 151M | 22.89 | - |
|  | - | 0 | 0 | - | 384 | 151M | 23.82 | - |
|  | - | 0 | 0 | - | 640 | 151M | 23.09† | 24.0† |
| LSH Transformer | - | 10 | 16 | Rand. Linear | 384 | 153M | 24.64 | - |
|  | - | 5 | 16 | Rand. Linear | 384 | 153M | 23.51 | - |
|  | - | 10 | 8 | Rand. Linear | 384 | 153M | 23.76 | - |
|  | - | 5 | 8 | Rand. Linear | 384 | 153M | 23.53 | - |
| LHA Transformer | (a) | 10 | 16 | Rand. | 384 | 153M | 26.05 | - |
|  | (b) | 10 | 16 | Linear | 384 | 153M | 24.22 | - |
|  | (c) | 10 | 16 | MLP | 384 | 153M | 24.00 | - |
|  | (d) | 5 | 16 | MLP | 384 | 153M | **22.79** | **23.2** |
|  | (e) | 10 | 8 | MLP | 384 | 153M | 23.00 | - |
|  | (f) | 5 | 8 | MLP | 384 | 153M | 22.84 | - |

Table 2: Performance of methods per task on the GLUE benchmark, where $C$ and $\#$ denote the number of queries/keys per cluster and hashing rounds, respectively. † denotes that the results are taken from their original papers.

|  | # | C | CoLA | MNLI | MRPC | QNLI | QQP | RTE | SST-2 | STS-B | AVG |
|---|---|---|---|---|---|---|---|---|---|---|---|
| RoBERTa-base | - | - | 60.9 | 87.6 | 88.7 | 92.7 | **91.6** | 68.5 | 94.5 | **90.0** | 84.3 |
| SMYRF† | 2 | 16 | 58.9 | 82.3 | 85.7 | 89.5 | 89.3 | 64.5 | 93.1 | 87.8 | 81.4 |
| SMYRF† | 2 | 32 | 58.8 | 85.0 | 87.7 | 91.1 | 89.7 | 68.6 | 93.2 | 89.7 | 83.0 |
| FCA† | - | - | 59.8 | 79.4 | 43.6 | 74.6 | 89.4 | 49.8 | 94.4 | 78.9 | 71.2 |
| i-FCA† | - | - | 60.1 | 88.0 | 87.3 | **93.0** | 91.5 | **70.4** | 94.7 | **90.0** | 84.4 |
| LSH | 1 | 11 | 61.5 | 86.3 | 88.7 | 91.1 | 90.2 | 68.0 | 93.3 | 88.5 | 83.5 |
| LSH | 1 | 22 | 61.5 | 86.8 | **89.7** | 91.9 | 90.8 | 68.4 | 93.9 | 88.7 | 84.0 |
| LSH | 1 | 45 | 61.8 | 87.5 | 88.7 | 92.7 | 91.2 | 68.1 | 94.0 | 89.4 | 84.2 |
| LHA | 1 | 11 | 61.2 | 85.7 | 89.1 | 91.3 | 90.5 | 68.0 | 93.7 | 89.0 | 83.6 |
| LHA | 1 | 22 | 62.0 | 86.8 | 88.0 | 92.4 | 91.1 | 68.6 | **94.8** | 89.6 | 84.2 |
| LHA | 1 | 45 | **62.3** | **87.7** | 89.1 | 92.8 | 91.4 | 68.5 | 94.6 | 89.5 | **84.5** |

per-layer sinusoidal absolute positional encoding (Vaswani et al., 2017). The LSH Transformer is implemented in our code base by simply replacing the learnable hash functions by random projections. A sequence length of 1536 is used in both training and evaluation stages for our models. In ablation study, we vary 1) the type of non-local attention heads, 2) the type of learnable hash function in LHA (i.e., linear or two-layer MLP), 3) the number of non-local attention heads, and 4) the number of non-local attention layers. We also consider a baseline where the hash functions generate content-independent random numbers. The experimental results are presented in Table 1.

From the table, we can see that a smaller attention size (384 compared to 768 in local Transformer) would decrease the model performance by 0.9, while adding LSH or LHA heads can improve the performance by 0.3 and 1.0, respectively. Surprisingly, we find that local Transformer is a very strong baseline for language modeling that outperforms a pure LHA Transformer. We can also see that a two-layer MLP works better than a linear function as the learnable hash functions in LHA. The best performance is achieved when we set half of the heads to local attention and the other half to non-local LHA for each layer. Finally, we can see that LHA consistently underperforms the untrainable LSH counterparts under four different settings. For the test set evaluation, we use the best setting from the ablation study. We can see that the proposed LHA model outperforms Transformer-XL, which further validate that LHA can help modeling long-term dependency for language modeling tasks.

## 5.2 NATURAL LANGUAGE UNDERSTANDING

To show the ability of our model to approximate arbitrarily complicated attention distributions, we evaluate our proposed method on the approximation of RoBERTa model (Liu et al., 2019) on the

Table 3: Accuracy on LO, IMDb, AAN, and Image in Long Range Arena benchmark. Best and second best model per task is shown in boldface and underlined. Throughput is evaluated on IMDb and relative to the vanilla transformer's. * and † denote being statistically significantly better ($p < 0.05$ in one-tail proportion test) than vanilla Transformer and the second best model. ‡ denotes that the throughput comparison are run on a single NVIDIA Tesla V100 32GB GPU, while previous results (Tay et al., 2020c) are reported on $4 \times 4$ TPU V3 chips. ♡, ◇, and ♠ denotes low-rank/kernelized attention, content-based sparse attention, and location-based sparse attention, respectively.

| Model | Accuracy (↑) | | | | | Throughput (↑) | | | |
|---|---|---|---|---|---|---|---|---|---|
| | LO | IMDb | AAN | Image | Avg. | 1K | 2K | 3K | 4K |
| Vaswani et al. (2017) - Transformer | 36.4 | 64.3 | 57.5 | 42.4 | 50.2 | 1.0 | 1.0 | 1.0 | 1.0 |
| Wang et al. (2020b) - Linformer♡ | 35.7 | 53.9 | 52.3 | 38.6 | 45.1 | 1.2 | 1.9 | 3.7 | 5.5 |
| Kitaev et al. (2020) - Reformer◇ | 37.3 | 56.1 | 53.4 | 38.1 | 46.2 | 0.5 | 0.4 | 0.7 | 0.8 |
| Child et al. (2019) - Sparse Trans.♠ | 17.1 | 63.6 | 59.6 | 44.2 | 46.1 | - | - | - | - |
| Tay et al. (2020b) - Sinkhorn Trans.◇ | 33.7 | 61.2 | 53.8 | 41.2 | 47.5 | 1.1 | 1.6 | 2.9 | 3.8 |
| Tay et al. (2020a) - Synthesizer♠ | 37.0 | 61.7 | 54.7 | 41.6 | 48.8 | 1.1 | 1.2 | 2.9 | 1.4 |
| Zaheer et al. (2020) - BigBird♠ | 36.0 | 64.0 | 59.3 | 40.8 | 50.0 | 0.9 | 0.8 | 1.2 | 1.1 |
| Katharopoulos et al. (2020) - Linear Trans.♡ | 16.1 | 65.9 | 53.1 | 42.3 | 44.4 | 1.1 | **1.9** | 3.7 | 5.6 |
| Choromanski et al. (2020) - Performer♡ | 18.0 | 65.4 | 53.8 | 42.8 | 45.0 | **1.2** | 1.9 | **3.8** | **5.7** |
| Peng et al. (2021) - RFA♡ | 36.8 | 66.0 | 56.1 | - | - | 1.1 | 1.7 | 3.4 | 5.3 |
| Ours - LHA◇ | **37.9** | **66.8**\*† | **62.8**\*† | **45.2**\* | **53.1** | 1.0‡ | 1.1‡ | 1.3‡ | 1.5‡ |

GLUE (General Language Understanding Evaluation) dataset (Wang et al., 2018). Following the common practice, the maximum sequence length is set to 128. To show the effectiveness of LHA, we choose two competitive baselines in literature: SMYRF (Daras et al., 2020) and Fast Clustered Attention (FCA) (Vyas et al., 2020). We also produce an LSH-based sparse attention baseline. We use a pure LHA model similar to variant (c) in Section 5.1.

We summarize the performance per task in Table 2. We report accuracy for all tasks except STS-B, where we report Pearson correlation. From the table, we can see that LHA performs as good as full attention for all the GLUE tasks, and that LHA outperforms all other methods in the average GLUE score, and has a smaller computational costs in its sparse attention part.

## 5.3 LONG-RANGE-ARENA BENCHMARK

Long-Range-Arena (LRA) benchmark (Tay et al., 2020c) is a recently proposed benchmark focused on evaluating model quality under long-context scenarios for Transformer variants. We follow the apples-to-apples setting[5] of LRA benchmark and compare our method against other efficient attention variants. We use an LHA/local hybrid variant similar to variant (d) in Section 5.1.

We consider the tasks of ListOps (Nangia & Bowman, 2018) (LO), byte-level IMDb reviews text classification (Maas et al., 2011) (IMDb), byte-level document retrieval on ACL Anthology Network (AAN) (Radev et al., 2013), and CIFAR10 (Krizhevsky et al., 2009) image classification on sequences of pixels (Image). The results are shown in Table 3 and the brief descriptions of the compared baselines can be found in Section 2. From the table, we can see that LHA achieves consistent improvements over previous efficient attention models.

## 6 CONCLUSION & FUTURE WORK

In this paper, we address the limitations of ANN-based sparse attention methods and propose the Learning-to-Hash Attention (LHA) as our new solution. Specifically, LHA leverages separate learnable hash functions for queries and keys, respectively, and utilizes kernelized techniques for efficient approximation of attention utilities. The experiments on language modeling, natural language understanding, text classification, and image classification demonstrated the effectiveness of LHA. For future work, we would like to validate the effectiveness of LHA model on much larger language modeling datasets, such as `PG-19` (Rae et al., 2019).

---

[5] `www.github.com/google-research/long-range-arena#apples-to-apples-setting`

## ACKNOWLEDGEMENT

This research used Perlmutter supercomputer of the National Energy Research Scientific Computing Center, a DOE Office of Science User Facility supported by the Office of Science of the U.S. Department of Energy under Contract No. DE-AC02-05CH11231 using NERSC award NERSC DDR-ERCAP0022110.

## ETHICS STATEMENT

This work proposed a more effective sparsification pattern for the attention mechanisms. We believe our proposed model as well as the code to be released can most benefit the field of language processing, with the potential to benefit other fields involving long sequence modeling. However, it is know that Transformers are also vulnerable to the adversarial attacks (Michel et al., 2019). Since one of the advantage of our model is to re-use a pre-trained Transformer model, the effective adversarial perturbations on the pre-trained model could also be effective in our model.

## REPRODUCIBILITY STATEMENT

- We provide a GitHub repository[6] for our source code. The hyper-parameters are described in details in the appendix. We also provide a pseudo-code implementaion of our model in the appendix.

- All the datasets used in the experiments and the corresponding pre-processing scripts can be found online, including language modeling[7], GLUE benchmark[8], Long-Range-Arena benchmark[9], and time series forecasting[10]. The pre-trained model we used (i.e., RoBERTa) is also publically available[11].

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

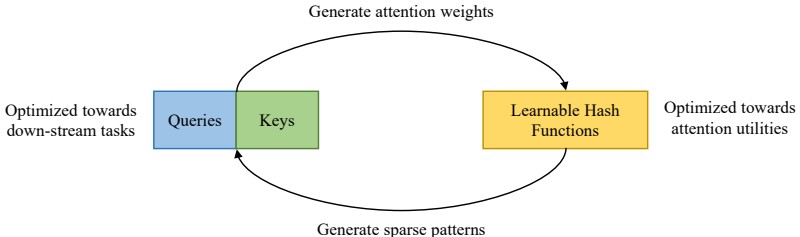

Figure 3: The joint training diagram of LHA, where queries and keys generate attention weights to train the learnable hash functions, while the hash functions generate the sparse attention patterns, on which queries and keys are trained towards down-stream tasks.

## A EXPERIMENTAL SETTINGS

### A.1 RE-EXAMINATION OF CONTENT-BASED SPARSE ATTENTION

For analysis, we pre-trained a 16-layer and 10-head Transformer on the WikiText-103 language modeling benchmark (Merity et al., 2016). A per-layer sinusoidal absolute positional encoding (Vaswani et al., 2017) is injected to the queries and keys before self-attention.

For Figure 1, we analyzed the queries and keys for the first 1024 tokens in the WikiText-103 validation data, and set the number of LSH buckets $B = 4$. We apply LSH to all the 10 attention heads in the $3^{rd}$ Transformer layer. For Figure 2, we follow the same experimental setting.

### A.2 LEARNING-TO-HASH

During the training stage, our final optimization objective $\mathcal{L}$ is a convex combination of the task-specific objective $L_{task}$ (e.g., cross-entropy loss in language modeling) and learning-to-hash objectives for all keys and queries, i.e., $\{\mathcal{L}_{Q_i}\}$ and $\{\mathcal{L}_{K_j}\}$ (See equation 13).

$$\mathcal{L} = (1-\lambda)\mathcal{L}_{task} + \frac{\lambda}{N \cdot L \cdot H} \sum_{i=1}^{N} \sum_{h=1}^{L} \sum_{h=1}^{H} (\mathcal{L}_{Q_i}^{L,H} + \mathcal{L}_{K_j}^{L,H}) \tag{19}$$

where $\lambda$, $N$, $L$, $H$ denote the loss coefficient, the sequence length, the number of attention layers, and the number of attention heads per layer, respectively. Empirically, we set $\Lambda = 0.05$ and found it work well across different tasks.

### A.3 LANGUAGE MODELING

We use the `base` setting of Transformer-XL (Dai et al., 2019) in our experiments, which consists of 16 Transformer layers. For each layer, the hidden size is set to 410, and the number of attention heads is set to 10. The dimension of feed-forward layer is set to 2100. All codes are implemented based on Flax (Heek et al., 2020) in JAX (Bradbury et al., 2018). The dropout ratio is set to 0.2. The batch size is set to 32. We use AdamW (Loshchilov & Hutter, 2017) as the optimizer, and set $(\beta_1, \beta_2)$ to (0.9, 0.999). The peak learning rate is set to 3.5e-4. The model is trained for 20k steps with a 2k-step warm-up stage with a cosine learning rate decay (Loshchilov & Hutter, 2016).

### A.4 NATURAL LANGUAGE UNDERSTANDING

We use the `base` setting of RoBERTa (Liu et al., 2019) in our experiments, which consists of 12 Transformer layers. For each layer, the hidden size is set to 768, and the number of attention heads is set to 12. The dimension of feed-forward layer is set to 3072. The RoBERTa pre-trained checkpoint is taken from the Transformers (Wolf et al., 2020) library. We use AdamW (Loshchilov & Hutter, 2017) as the optimizer, and set $(\beta_1, \beta_2)$ to (0.9, 0.98). To fine-tune the pre-trained models, we search the optimization hyper-parameters in a search space including different batch sizes (16/32/48), learning rates ((1-5) * 1e-5), and the number of epochs (3-5). A $10\%$ warm-up schedule and linear learning rate decay (Devlin et al., 2018) is applied.

Table 4: Univariate long sequence time series forecasting results on two datasets (lower is better). †denotes that the results are taken from (Zhou et al., 2020).

| Models | | Ours - LHA | | Informer† | | LogTrans† | | Reformer† | | DeepAR† | | Prophet† | |
|---|---|---|---|---|---|---|---|---|---|---|---|---|---|
| Metric | | MSE | MAE | MSE | MAE | MSE | MAE | MSE | MAE | MSE | MAE | MSE | MAE |
| ETTh₁ | 24 | **0.060** | **0.191** | 0.098 | 0.247 | 0.103 | 0.259 | 0.222 | 0.389 | 0.107 | 0.280 | 0.115 | 0.275 |
| | 48 | **0.088** | **0.236** | 0.158 | 0.319 | 0.161 | 0.322 | 0.284 | 0.445 | 0.162 | 0.327 | 0.168 | 0.330 |
| | 168 | **0.158** | **0.328** | 0.183 | 0.346 | 0.187 | 0.355 | 1.522 | 1.191 | 0.239 | 0.422 | 1.224 | 0.763 |
| | 336 | **0.219** | 0.399 | 0.222 | **0.387** | 0.230 | 0.398 | 1.860 | 1.124 | 0.445 | 0.552 | 1.549 | 1.820 |
| | 720 | **0.265** | 0.437 | 0.269 | **0.435** | 0.273 | 0.463 | 2.112 | 1.436 | 0.658 | 0.707 | 2.735 | 3.253 |
| ETTm₁ | 24 | **0.016** | **0.102** | 0.030 | 0.137 | 0.065 | 0.202 | 0.095 | 0.228 | 0.091 | 0.243 | 0.120 | 0.290 |
| | 48 | **0.021** | **0.112** | 0.069 | 0.203 | 0.078 | 0.220 | 0.249 | 0.390 | 0.219 | 0.362 | 0.133 | 0.305 |
| | 96 | **0.054** | **0.180** | 0.194 | 0.372 | 0.199 | 0.386 | 0.920 | 0.767 | 0.364 | 0.496 | 0.194 | 0.396 |
| | 288 | **0.121** | **0.277** | 0.401 | 0.554 | 0.411 | 0.572 | 1.108 | 1.245 | 0.948 | 0.795 | 0.452 | 0.574 |
| | 672 | **0.403** | **0.572** | 0.512 | 0.644 | 0.598 | 0.702 | 1.793 | 1.528 | 2.437 | 1.352 | 2.747 | 1.174 |

## A.5 LONG-RANGE-ARENA BENCHMARK

We follow the apples-to apples setting[12] of LRA benchmark and compare our method against other efficient attention variants. Specifically, we use a 4-layer Transformer with 256 hidden size and 1024 feed-forward layer size for the IMDb text classification task(Maas et al., 2011), a 4-layer Transformer with 256 hidden size and 1024 feed-forward layer size for the ListOps task(Radev et al., 2013), a 4-layer Transformer with 128 hidden size and 512 feed-forward layer size for the document retrieval task(Nangia & Bowman, 2018), and a 4-layer Transformer with 128 hidden size and 64 feed-forward layer size for the CIFAR10 image classification task(Krizhevsky et al., 2009). The models are trained for 20k steps with a 2k-step warm-up stage with a cosine learning rate decay (Loshchilov & Hutter, 2016).

## A.6 TIME SERIES FORECASTING

We follow the experimental setup of Informer (Zhou et al., 2020). Specifically, the input length of recurrent component is chosen from $\{24, 48, 96, 168, 336, 720\}$ for the ETTh1, and chosen from $\{24, 48, 96, 192, 288, 672\}$ for the ETTm dataset. The layer of encoder is chosen from $\{6, 4, 3, 2\}$ and the layer of decoder is set as 2. The head number of multi-head attention is chosen from $\{8, 16\}$, and the dimension of multi-head attention's output is set as 512.

## A.7 COMPUTING INFRASTRUCTURE

All the model training are conducted on a machine with 4 NVIDIA Ampere A100 40GB GPUs and 64 AMD EPYC 7713 64-Core Processor in a Slurm (Yoo et al., 2003) system. The evaluation of the inference throughput is performed on a stand-alone machine with 1 NVIDIA Tesla V100 32GB GPU.

## B ADDITIONAL EXPERIMENTS ON TIME SERIES FORECASTING

We also evaluate our model on time series forecasting tasks. We use the ETT (Electricity Transformer Temperature) dataset (Zhou et al., 2020), which contains 2-year data from two separated counties in China. ETTh₁ is a dataset for 1-hour-level, while ETTm₁ is a dataset for 15-minute-level. Each data point consists of the target value "oil temperature" and 6 power load features. The train/val/test is 12/4/4 months. The Mean Squared Error (MSE) metric and Mean Average Error (MAE) metric are used as the evaluation metrics. We use an LHA/local hybrid variant similar to variant (d) in Section 5.1.

In univariate forecasting setting, all the seven features are used as input and "oil temperature" is the prediction target. The univariate evaluation results can be found in Table 4. We can see that LHA significantly improve the performance of state-of-the-art for most settings, while slightly underperforms Informer (Zhou et al., 2020) in two settings of ETTh₁ dataset. This verifies the effectiveness of LHA on time series data.

---

[12]`www.github.com/google-research/long-range-arena#apples-to-apples-setting`

Table 5: Multivariate long sequence time series forecasting results on two datasets (lower is better). †denotes that the results are taken from (Zhou et al., 2020).

| Models | Ours - LHA | | Informer† | | LogTrans† | | Reformer† | | LSTnet† | |
|---|---|---|---|---|---|---|---|---|---|---|
| Metric | MSE | MAE | MSE | MAE | MSE | MAE | MSE | MAE | MSE | MAE |
| ETTh₁ 24 | **0.475** | **0.469** | 0.577 | 0.549 | 0.686 | 0.604 | 0.991 | 0.754 | 1.293 | 0.901 |
| 48 | **0.569** | **0.544** | 0.685 | 0.625 | 0.766 | 0.757 | 1.313 | 0.906 | 1.456 | 0.960 |
| 168 | **0.883** | **0.706** | 0.931 | 0.752 | 1.002 | 0.846 | 1.824 | 1.138 | 1.997 | 1.214 |
| 336 | **0.970** | **0.793** | 1.128 | 0.873 | 1.362 | 0.952 | 2.117 | 1.280 | 2.655 | 1.369 |
| 720 | **1.021** | **0.805** | 1.215 | 0.896 | 1.397 | 1.291 | 2.415 | 1.520 | 2.143 | 1.380 |
| ETTm₁ 24 | **0.056** | **0.191** | 0.323 | 0.369 | 0.419 | 0.412 | 0.724 | 0.607 | 1.968 | 1.170 |
| 48 | **0.088** | **0.236** | 0.494 | 0.503 | 0.507 | 0.583 | 1.098 | 0.777 | 1.999 | 1.215 |
| 96 | **0.158** | **0.328** | 0.678 | 0.614 | 0.768 | 0.792 | 1.433 | 0.945 | 2.762 | 1.542 |
| 288 | **0.219** | **0.399** | 1.056 | 0.786 | 1.462 | 1.320 | 1.820 | 1.094 | 1.257 | 2.076 |
| 672 | **0.265** | **0.437** | 1.192 | 0.926 | 1.669 | 1.461 | 2.187 | 1.232 | 1.917 | 2.941 |

In multivariate forecasting setting, all the seven features are the prediction targets. The multivariate evaluation results can be found in Table 5. We can see that LHA significantly improve the performance of state-of-the-art for all settings, which further demonstrates the effectiveness of the proposed LHA method.

## C  PROOF

We follow the notations in (Daras et al., 2020). Let $\mathcal{C}^B$ the set of all possible assignments in $B$ balanced non-overlapping clusters. A specific assignment is denoted by $\mathcal{C}_t^B$ and there are $T$ possible such assignments:

$$\mathcal{C}^L = \{\mathcal{C}_1^B, C_2^B, ... \mathcal{C}_T^B\}. \tag{20}$$

$$\mathcal{C}_t^B = \{c_1, c_2, ..., c_B\}: \quad \begin{cases} c_i = \{Q_1^i, ..., Q_{\frac{N}{B}}^i, K_1^i, ..., K_{\frac{N}{B}}^i\} & c_i \subseteq \mathcal{Q} \cup \mathcal{K}, \ \forall i \in \{1, ..., L\} \\ c_x \cap c_y = \varnothing & \forall c_x, c_y \in \mathcal{C}_t^B \end{cases} \tag{21}$$

We have the following lemma, which is referred as the max-mass problem:

**Lemma 1.** *(Daras et al., 2020) The optimization problem:*

$$\max_{\mathcal{C}_t^B \in \mathcal{C}^B} \sum_{(Q_i, K_j) \in \mathcal{C}_t^B} Q_i \cdot K_j \tag{22}$$

*is NP-hard.*

The main idea of the proof is to show that solving polynomially the above problem would mean that we could also solve in polynomial time the 3-DM, which is known to be NP-complete. Please refer to (Daras et al., 2020) for the detailed constructive proofs. Next, we show how to use this lemma to prove Theorem 1.

*Proof of Theorem 1.* Let $\{Q_i\}$ and $\{K_j\}$ denote the query and key sets that we consider for computing the attention utility. We can first construct the new query set $\mathcal{Q}'$ and key set $\mathcal{K}'$ such that

$$Q_i' \cdot K_j' = \frac{e^{Q_i \cdot K_j}}{\sqrt{d_h}}, \quad \text{where} \quad Q_i' \in \mathcal{Q}', K_j' \in \mathcal{K}'$$

This can be achieved by applying SVD to the matrix of $\left(\frac{e^{Q_i \cdot K_j}}{\sqrt{d_h}}\right)_{ij}$. Next, we construct another query set $\mathcal{Q}''$, such that

$$Q_i'' = \frac{Q_i'}{\sum_{j=1}^n Q_i' \cdot K_j'}$$

The problem of finding the optimal attention utility, i.e.,

$$\arg\max_{\mathcal{C}_t \in \mathcal{C}} \text{AU}(\mathcal{C}_t) = \sum_{(Q_i, K_j) \in \mathcal{C}_t} A_{ij}$$

is thus equivalent to the problem in Lemma 1, with $\mathcal{Q}''$ and $\mathcal{K}'$ as the query and key sets, which is proven to be NP-hard. $\qquad\qquad\qquad\qquad\qquad\qquad\qquad\qquad\qquad\qquad\qquad\qquad\qquad\qquad\qquad\qquad\square$

## D  ALGORITHM

We present a detailed pseudo-code implementation for LHA in Algorithm 1.

---

**Algorithm 1** Single-layer Single-head Learning-to-Hash Attention (LHA)

---

1: Queries, Keys and Values: $Q, K, V \in \mathbb{R}^{N \times d_h}$
2: Query/Key Hash Functions: $H_Q, H_K$
3: Attention Utility Loss Weight: $\lambda$
4: **if** causal mask **then**
5:      $Q_{score} \leftarrow \text{softmax}(H_Q(Q) + H_K(K))$                              $\triangleright B \times N \times d_h$
6:      $K_{score} \leftarrow Q_{score}$
7: **else**
8:      $Q_{score} \leftarrow \text{softmax}(H_Q(Q))$                                        $\triangleright B \times N \times d_h$
9:      $K_{score} \leftarrow \text{softmax}(H_K(K))$                                   $\triangleright B \times N \times d_h$
10: $W \leftarrow N/B$                                                     $\triangleright$ bucket size
11: $Q_{idx} \leftarrow \text{top-}\{\sqrt{2}\text{W}\}(Q_{score})$                           $\triangleright B \times N \log(N)$
12: $K_{idx} \leftarrow \text{top-}\{\sqrt{2}\text{W}\}(K_{score})$                          $\triangleright B \times N \log(N)$
13: $Q' \leftarrow \text{gather}(Q, Q_{idx})$                               $\triangleright B \times W \times d_h$
14: $K' \leftarrow \text{gather}(K, K_{idx})$                               $\triangleright B \times W \times d_h$
15: $V' \leftarrow \text{gather}(V, K_{idx})$                                $\triangleright B \times W \times d_h$
16: $A \leftarrow Q'(K')^T$                                   $\triangleright B \times W \times W \times d_h$
17: **if** causal mask **then**
18:      $A \leftarrow \text{mask}(A)$                                     $\triangleright B \times W \times W$
19: $A \leftarrow \text{softmax}(A)$.                                  $\triangleright B \times W \times W$
20: $V' \leftarrow A \cdot V'$                                     $\triangleright B \times W \times W \times d_h$
21: $X \leftarrow \text{merge}(K_{idx}, V')$                             $\triangleright B \times W \times d_h$
22: $\boldsymbol{\psi} = \text{AU}_Q(Q, K)$                               $\triangleright B \times N$, Equation 16
23: $\boldsymbol{\psi}' = \text{AU}_K(Q, K)$                              $\triangleright B \times N$, Equation 17
24: $\mathcal{L}_{AU} = \lambda \cdot KL(\boldsymbol{\psi} \| Q_{score}) + \lambda \cdot KL(\boldsymbol{\psi}' \| K_{score})$
25: **return** $X, \mathcal{L}_{AU}$

---

## E  A COMPLETE DERIVATION OF THE LHA OBJECTIVE

Let $Q, K, V \in \mathbb{R}^{N \times d_h}$ denote the query, key, and value vectors of the attention mechanism, where $N$ is the sequence length and $d_h$ is the hidden size. Let $\phi : \mathbb{R}^{d_h} \mapsto \mathbb{R}^{2D}$ be Positive Randomized Features (PRFs) (Choromanski et al., 2020) such that:

$$\mathbb{E}[\phi(\mathbf{x}) \cdot \phi(\mathbf{y})] = \exp\left(\frac{\mathbf{x}^T \mathbf{y}}{\sqrt{d_h}}\right)$$

We can approximate the aggregated attention utility (Eq. 5) by:

$$\boldsymbol{\psi}_b^{(i)} = \sum_{j:h_K(K_j)=b} A_{ij} \propto \sum_{j:h_K(K_j)=b} \exp\left(\frac{Q_i \cdot K_j}{\sqrt{d_h}}\right) \propto \mathbb{E}\left[\phi(Q_i) \cdot \sum_{j:h_K(K_j)=b} \phi(K_j)\right]$$

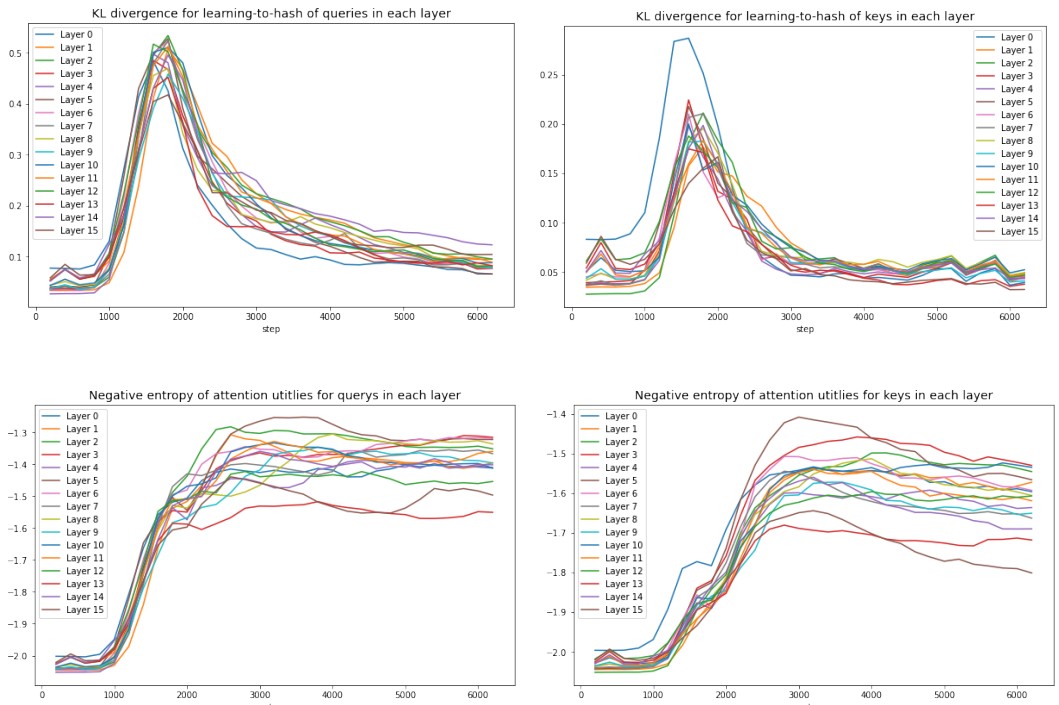

Figure 4: The training dynamics of the KL divergence and negative entropy in LHA for queries and keys.

$$\psi_b'^{(i)} = \sum_{i:h_Q(Q_i)=b} A_{ij} = \frac{\sum_{j:h_Q(Q_i)=b} \exp\left(\frac{Q_i \cdot K_j}{\sqrt{d_h}}\right)}{\sum_{j=1}^{N} \exp\left(\frac{Q_i \cdot K_j}{\sqrt{d_h}}\right)}$$

$$\propto \frac{\mathbb{E}\left[\phi(Q_i) \cdot \sum_{j:h_K(K_j)=b} \phi(K_j)\right]}{\mathbb{E}\left[\phi(Q_i) \cdot \sum_{j=1}^{N} \phi(K_j)\right]}$$

$$\asymp \mathbb{E}\left[\frac{\phi(Q_i) \cdot \sum_{j:h_K(K_j)=b} \phi(K_j)}{\phi(Q_i) \cdot \sum_{j=1}^{N} \phi(K_j)}\right]$$

$$\propto \mathbb{E}\left[\left(\sum_{j:h_Q(Q_i)=b} \frac{\phi(Q_i)}{\phi(Q_i) \cdot \sum_{j=1}^{N} \phi(K_j)}\right) \cdot \phi(K_j)\right]$$

where $\asymp$ denotes that the attention approximation converges under the positive random feature condition (Katharopoulos et al., 2020). Next, the Learning-to-Hash objectives for each query and each key can be defined as:

$$\mathcal{L}_{Q_i} = KL(\psi^{(i)} \| \text{softmax}(H_Q(Q_i))) \tag{23}$$

$$\mathcal{L}_{K_i} = KL(\psi'^{(i)} \| \text{softmax}(H_K(K_i))) \tag{24}$$

The final optimization objective $\mathcal{L}$ is a convex combination of the task-specific objective $L_{task}$ (e.g., cross-entropy loss in language modeling) and learning-to-hash objectives for all keys and queries in each attention head for each layer:

$$\mathcal{L} = (1-\lambda)\mathcal{L}_{task} + \frac{\lambda}{N \cdot L \cdot H} \sum_{i=1}^{N} \sum_{h=1}^{L} \sum_{h=1}^{H} (\mathcal{L}_{Q_i}^{L,H} + \mathcal{L}_{K_j}^{L,H}) \tag{25}$$

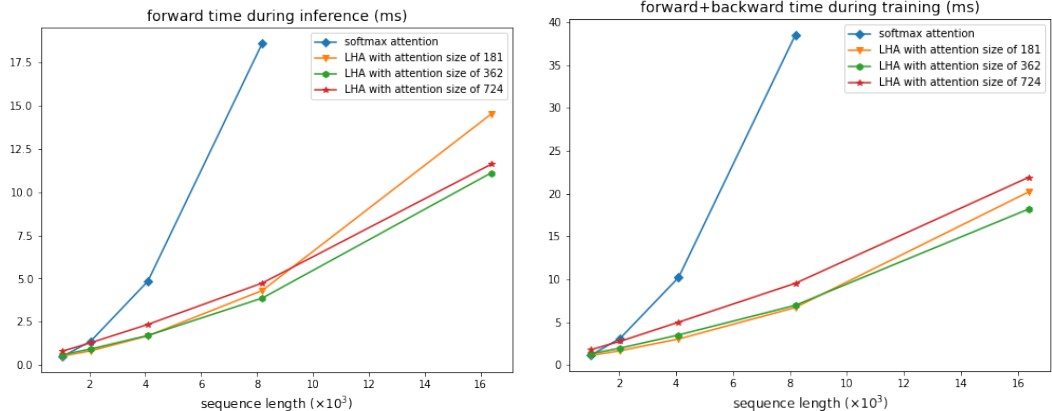

Figure 5: The training and inference latency of LHA and softmax attention. The latency is measured on a single NVIDIA Tesla A100 GPU with a batch size of 1.

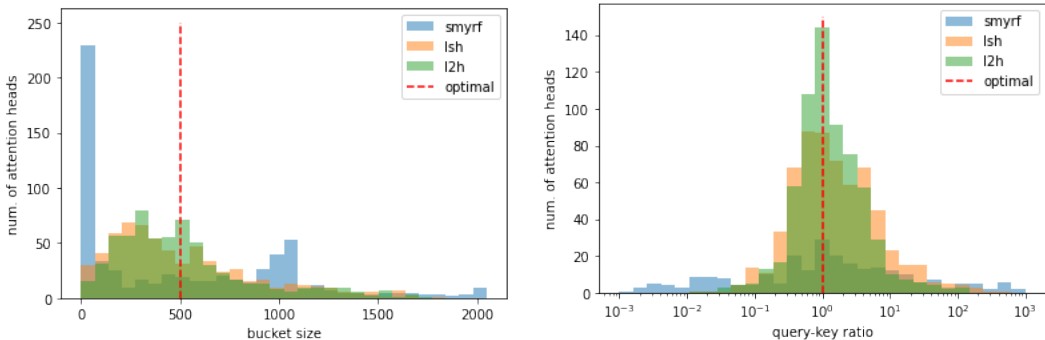

Figure 6: We show the histogram of hash bucket statistics for all $16 \times 10 = 160$ attention heads in the a pre-trained 16-layer Transformer. For each head, we assign the queries and keys of first 1024 tokens in the WikiText-103 validation data into 4 LSH buckets. (left) The histogram of the bucket sizes for all $160 \times 4 = 640$ buckets. (right) The histogram of the query-key ratios for all $160 \times 4 = 640$ buckets. The dashed red lines denote the optimal case where each hash bucket has the same size and same number of queries and keys.

## F    TRAINING DYNAMICS

We plot the training dynamics of the KL divergence (i.e., $\mathcal{L}_Q$ and $\mathcal{L}_K$) and the negative entropy of the bucket-wise attention utilities (i.e., negative entropy of $\psi$ for queries and $\psi'$ for keys) in Figure 4. KL divergence is the optimization objective of LHA, while negative entropy measures the diversity of bucket-wise attention utility. The model we use is a pure LHA model with 16 layers and 10 LHA heads, trained on WikiText-103 for 6000 steps. Each LHA head has 8 hash buckets. From the plots, we can see that the values of negative entropy consistently increase during training for both queries and keys, while the values of KL divergence first increase (due to the increase of the diversity of bucket-wise attention utility), and then decrease (due to learning-to-hash optimization).

## G    ADDITIONAL ANALYSIS ON TRAINING & INFERENCE EFFICIENCY

To further analyze the efficiency of the proposed LHA compared to softmax attention, we measure the latency of LHA and softmax attention for both training and inference stage, varying by sequence length. Notice that in the inference stage, the LHA model needs not to calculate the learning-to-hash losses (i.e., the KL divergence terms). We study a single-layer attention, which has 10 attention heads and the hidden size of 410. Figure 5 illustrates the latency in both training and inference stage for

softmax attention and LHA with different attention size. We can see that LHA can achieve more significant speedup when facing longer sequences.

# H ADDITIONAL ANALYSIS ON THE BUCKET IMBALANCE ISSUES

To further analyze how the bucket imbalance issues are alleviated by the proposed LHA, we plot the histograms of bucket sizes and query-key ratios for all $16 \times 10 = 160$ attention heads in a pre-trained Transformer. The results are shown in Figure 6. We can see that learning-to-hash attention (l2h) has more buckets close to the optimal bucket size (i.e., 512) and has more balanced query-key ratios for hash buckets.

Notice that to compare the statistics of the bucket size and query-key ratios between LSH and LHA, we fine-tune learnable hash functions for the same pre-trained Transformer. Notice that in this experiment, we directly use the highest-ranked bucket as the hash bucket for each query and key, instead of using a token sorting strategy (Roy et al. 2021) to maintain the same bucket sizes. When calculating the query-key ratios, we do not count the buckets which have no queries or keys.

