# OpenReview forum: "Sparse Attention with Learning to Hash"
_ICLR.cc/2022/Conference — ICLR 2022 Poster_

### Official Review · Reviewer_wDdo · 2021-11-01

**Correctness:** 3
**Technical Novelty And Significance:** 2
**Empirical Novelty And Significance:** 3
**Recommendation:** 5
**Confidence:** 5

**Details Of Ethics Concerns:**

See the above Main Review Section.

**Main Review:**

The authors figure the bucket imbalance problem in ANN-derived content-based sparse attention and analyze its weakness on the imbalance problem. Attention Biclusteringis introduced to find the optimal attention utility. The learning to hash-based attention model is proposed to improve the effectiveness of sparse attention. Experiments on different applications support their claims.

Concerns:
1. It is well-known that LSH is data-independent hashing, which is formulated by random projection with some pre-defined metrics. Notably, using learning to hash is better than the LSH and its variants, in most cases, for representation learning. Therefore, the authors need to clarify why replacing LSH with learning to hash models is useful in sparse attention.
2. Such a simple replacement could fully support your work in the present form. The authors should give full reasons for your contributions.
3. The authors fail to convince the reviewer what are the connections between Theorem 1 and your proposed method. To me, you can directly claim there are some limitations on LSH-based methods, and implementing learning to hash can improve its representation capabilities on feature learning.
4. Efficiency is one of the proposed methods, and more analysis on the efficiency and efficacy is necessary.


**Summary Of The Paper:**

This work studies the content-based sparse attention in transformer and how to improve the self-attention part, i.e., efficiency and effectiveness.

**Summary Of The Review:**

The authors try to improve some limitaions on the content-based sparse attention when using LSH-produced sparse attention patterns. A Learning-to-Hash Attention is formulated to enhance its model expressiveness. Experiments show its usefulness of the proposed methods. However, there are questionable points that should be clarified.

---

> ### Author Response · Authors · 2021-11-15
> **Response to Reviewer wDdo**
>
> We greatly appreciate the review, especially the concerns on efficiency analysis.
>
> > “It is well-known that LSH is data-independent hashing, which is formulated by random projection with some pre-defined metrics. Notably, using learning to hash is better than the LSH and its variants, in most cases, for representation learning. Therefore, the authors need to clarify why replacing LSH with learning to hash models is useful in sparse attention.”
>
> > “Such a simple replacement could fully support your work in the present form. The authors should give full reasons for your contributions.”
>
> Apparently, the reviewer agrees with our point that data-dependent learning-to-hash (L2H) is a better alternative than data-independent LSH in general.  However, how to apply L2H for content-based sparse attention sparsification in Transformer-based models has not been explored so far. Addressing this challenge is the novel contribution of our paper. That is, we are the first to show that learning-to-hash leads to better attention sparsification patterns.
>
> > “The authors fail to convince the reviewer what are the connections between Theorem 1 and your proposed method. To me, you can directly claim there are some limitations on LSH-based methods, and implementing learning to hash can improve its representation capabilities on feature learning.”
>
> We have analyzed intensively in this paper the limitations of LSH-based methods. Nevertheless, it is still meaningful to show Theorem 1, i.e., the NP-hardness of maximizing attention utility to give a conceptually clear big picture.  In the rest of the paper, we offer a computationally tractable formulation with efficient algorithms for solving this problem with learning-to-hash.
>
> > “Efficiency is one of the proposed methods, and more analysis on the efficiency and efficacy is necessary.”
>
> We thank the reviewer for the suggestion. We have added an additional efficiency analysis of LHA in Section G of the appendix. Our results show that LHA can achieve more significant speedup when facing longer sequences. The efficacy of the proposed method is verified with extensive experimental results in the paper.

---

> ### Author Response · Authors · 2021-11-29
> **Response to Reviewer wDdo (Follow-up)**
>
> We greatly appreciate your great effort and time in reviewing our work and providing constructive feedback!
>
> We are following up to check whether our rebuttal responses have fully addressed your comments/concerns.
>
> Thank you, Authors

---

### Official Review · Reviewer_fhAP · 2021-11-02

**Correctness:** 3
**Technical Novelty And Significance:** 2
**Empirical Novelty And Significance:** 3
**Recommendation:** 6
**Confidence:** 3

**Main Review:**

### Strengths:
* The attention sparsification is important to reduce the complexity of Transformer when applied to long sequence.

* The imbalance of bucket size and query-key ratios are studied, which is closely related to efficiently reducing the complexity and performance on down-stream tasks.

* The attention utility is proposed as a metric to measure how well the attention weights are preserved.

* The attention utility is used to train the learnable hash functions since the argmax operation makes it impossible to train the hash functions from the down-stream tasks.

* An approximation of the attention utilities is proposed with random Fourier features to reduce the complexity.

### Weaknesses:

* What is the statistics of the bucket size and query-key ratios for the proposed LHA? Is it significantly better than LSH?

* It is not clear how to formulate the final training objective and balance the loss terms.

* It is not clear how to apply LHA as a plug-and-play replacement for dense attention layers in pre-trained Transformer models. Since the LHA method introduces the learnable hash functions h_k and h_Q, the hash functions should be trained on the target dataset.

* The implementation is inconsistent with the analyze. By using the token sorting method in Roy et al. 2021, the validation of attention bi-clustering is no longer guaranteed. However, the attention utility is meaningful only if the  attention bi-clustering is guaranteed.


### Minor:

* The notation of key and query is confused. Q_i should be the query, and a key should use K_i (before (14)).

* To make the paper self-contained, it is necessary to include a brief introduction to the compared baselines in Table 3. Adding a column to show what type of attention sparsification would help the reader to understand the comparison.

* Below(14): is not longer --> is no longer

* Below (16): h_K(K_j) does change --> does not change

**Summary Of The Paper:**

This paper proposes a learning to hash attention (LHA) to learn sparse attention for Transformer.
The proposed LHA addresses the limitation of ANN-based sparse attention method by separate learnable hash functions for queries and keys and utilizes kernelized techniques for efficient approximation of attention utilities.
Experiments on several applications validate the effectiveness of the proposed LHA.

**Summary Of The Review:**

This paper propose to learn separate hash functions for keys and queries with the guidance of attention utilization, which is the metric proposed in this paper. However, the implementation is inconsistent with the analyze and important method details are missing. At this point, I would tend to vote this paper slightly below the bar. If the authors can address my concerns, I would be happy to increase my score.

---

> ### Author Response · Authors · 2021-11-15
> **Response to Reviewer fhAP**
>
> We greatly appreciate the review, especially the questions on the implementation.
>
> > “What is the statistics of the bucket size and query-key ratios for the proposed LHA? Is it significantly better than LSH?”
>
> We thank the reviewer for raising this question. Inspired by the reviewer’s question, we conducted an additional analysis on the bucket imbalance issues in Section G of the appendix, where we plot the histograms of bucket sizes and query-key ratios for all $16 \times 10 = 160$ attention heads in a pre-trained Transformer. Our results show that We can see that LHA has more buckets close to the optimal bucket size and has more balanced query-key ratios, thus alleviating the bucket imbalance issues.
>
> > “It is not clear how to formulate the final training objective and balance the loss terms.”
>
> The final training objective is “a convex combination of the task-specific objective and the learning-to-hash objectives for all queries and keys” (page 6). The weight  $\lambda$ in the combination is tuned on validation data, as described in A.2 section of the appendix. The learning-to-hash objective is defined according to Equation 13. We also added a complete derivation of our LHA objective in Section E of the appendix.
>
> > “It is not clear how to apply LHA as a plug-and-play replacement for dense attention layers in pre-trained Transformer models. Since the LHA method introduces the learnable hash functions h_k and h_Q, the hash functions should be trained on the target dataset.”
>
> The plug-and-play replacement means that the proposed LHA method does not impose extra constraints for queries and keys, which allows it to be directly applied on pre-trained Transformer models, such as RoBERTa, without any retraining.
>
> Besides, many fast Transformer variants (e.g., Performer, RFA, BigBird) require a fine-tuning procedure (initialized from original Transformer) on the target dataset, due to the discrepancy between the full softmax attention and new attention functions. Taking this into consideration, the overhead introduced by training hash functions can be negligible.
>
> > “The implementation is inconsistent with the analyze. By using the token sorting method in Roy et al. 2021, the validation of attention bi-clustering is no longer guaranteed. However, the attention utility is meaningful only if the attention bi-clustering is guaranteed.”
>
> Let us clarify that finding the exact bi-clustering w.r.t. maximizing the aggregated attention utility (i.e., $\boldsymbol{\psi}$ and $\boldsymbol{\psi}'$) can be regarded as a Linear Assignment Problem (LAP), which is too costly on modern parallel-computing hardware. Thus, we use the token sorting method introduced by Roy et al. 2021 (with additional normalization) as an efficient approximation of the bi-clustering.  We should also point out that the attention utility is still meaningful if the attention is not strictly a bi-clustering. In this case, the attention utility represents the (normalized) aggregation of the sparse attention weights in each bucket.
>
> We have made the above points clear in the revised version of the paper.
>
> > “To make the paper self-contained, it is necessary to include a brief introduction to the compared baselines in Table 3. Adding a column to show what type of attention sparsification would help the reader to understand the comparison.”
>
> We thank the reviewer for the suggestion. The introduction to the compared baselines can be found in Section 2 Related Work. We have added symbols to show the type of attention in Table 3.

---

> > ### Comment · Reviewer_fhAP · 2021-11-24
> > **Response to Authors**
> >
> > Thanks the authors for the detailed response.
> > >We thank the reviewer for raising this question. Inspired by the reviewer’s question, we conducted an additional analysis on the bucket imbalance issues in Section G of the appendix, where we plot the histograms of bucket sizes and query-key ratios for all  attention heads in a pre-trained Transformer. Our results show that We can see that LHA has more buckets close to the optimal bucket size and has more balanced query-key ratios, thus alleviating the bucket imbalance issues.
> >
> > The results are quite interesting and convincing.
> >
> > >The final training objective is “a convex combination of the task-specific objective and the learning-to-hash objectives for all queries and keys” (page 6). The weight  in the combination is tuned on validation data, as described in A.2 section of the appendix. The learning-to-hash objective is defined according to Equation 13. We also added a complete derivation of our LHA objective in Section E of the appendix.
> >
> > The objective is now clear.
> >
> > >The plug-and-play replacement means that the proposed LHA method does not impose extra constraints for queries and keys, which allows it to be directly applied on pre-trained Transformer models, such as RoBERTa, without any retraining.
> > Besides, many fast Transformer variants (e.g., Performer, RFA, BigBird) require a fine-tuning procedure (initialized from original Transformer) on the target dataset, due to the discrepancy between the full softmax attention and new attention functions. Taking this into consideration, the overhead introduced by training hash functions can be negligible.
> >
> > I agree that many Transformer variants require a fine-tuning procedure and the overhead for training hash functions in this work can be negligible if jointly trained. However, it is important to point out the proposed hashing function requires additional learning effort since we are comparing with LSH based baselines. And we should do an additional learning step as compared to using LSH if we apply this method on a pre-trained Transformer model.
> >
> > >Let us clarify that finding the exact bi-clustering w.r.t. maximizing the aggregated attention utility (i.e.,  and ) can be regarded as a Linear Assignment Problem (LAP), which is too costly on modern parallel-computing hardware. Thus, we use the token sorting method introduced by Roy et al. 2021 (with additional normalization) as an efficient approximation of the bi-clustering. We should also point out that the attention utility is still meaningful if the attention is not strictly a bi-clustering. In this case, the attention utility represents the (normalized) aggregation of the sparse attention weights in each bucket.
> > We have made the above points clear in the revised version of the paper.
> >
> > It is now clear to me.
> >
> > >We thank the reviewer for the suggestion. The introduction to the compared baselines can be found in Section 2 Related Work. We have added symbols to show the type of attention in Table 3.
> >
> > Thanks for the revision.
> >
> > The authors' response and revision have almost addressed my concerns. I am happy to increase the rating to 6, but I think it is still a borderline paper. Thank you.

---

### Official Review · Reviewer_zEkG · 2021-11-07

**Correctness:** 3
**Technical Novelty And Significance:** 3
**Empirical Novelty And Significance:** 3
**Recommendation:** 8
**Confidence:** 4

**Main Review:**

Pros:

I really enjoyed reading this paper. It starts by departing itself from related works by investigating the critical issue of not having balanced buckets, showing that this characteristic is a common pitfall of previous popular works. Therefore, the foundation is clear and has practical implications.

Moreover, quantifying the degree to which an approximate self-attention can imitate the full attention is very informative and might guide future work on fast transformers. The authors acknowledge that computing this number is an NP-hard problem, but they go further and try to optimize this metric during training.

The idea of having a learnable function prior to bucketing is not entirely novel (see a contemporary idea by Treviso et al., 2021), but the formulation is concise and clear, enabling generalizations to previous approaches.

The experiments span a large set of tasks in NLP and CV with increasingly large input lengths. LM experiments show that the proposed method (trainable LSH) outperforms Reformer (untrainable LSH) on multiple settings. Results on the GLUE benchmark show that the proposed method achieves results on par with the original RoBERTa model, which is a good sanity check. The experiments on Long Range Arena give the final flavor to the paper, with the proposed method outperforming related works on all considered tasks while being cheaper (higher throughput) to train than a standard transformer.


Cons:

However, I was also able to spot some concerns in this paper. Concretely:

- The core of the proposed method relies on optimizing the attention utility, which relies on the Performer's softmax kernel to regularize the training regime. The paper opts to use a KL between the softmax kernel and LHA to that end. It would be interesting to see how this KL term evolves during training.

- The choice of 10 heads from the 3rd layer in section 3.2 looks arbitrary. In fact, it is widely known that different heads at different layers focus on distinct tokens. Thus, what is the motivation for that choice? Moreover, from which pre-trained model were the results extracted? This is not clear in the paper.

- Figure 1 shows the bucket size of queries and keys for each attention head using LSH attention. Since LSH attention was originally designed to use tied queries and keys (i.e., queries = keys), how can queries and keys have different bucket sizes? Also, it is unclear if queries and keys were tied for all experiments using LSH attention in the rest of the paper.

- Which setting regarding the number of attention heads and layers was used for experiments on the GLUE benchmark and LRA? That is, how many layers and heads were set to use LHA?

- In section 3.2, it is said that "queries should attend to enough keys to get a good approximation of the full-attention.". Is there a precise notion of what is "enough" in this context? That is, for recovering the true softmax distribution, queries should attend to ALL keys. In contrast, if we aim to recover an entmax distribution, queries can attend to only a subset of keys (see the sparse-consistency property in Treviso et al., 2021). So, what would change in LHA if we want to approximate entmax instead of softmax?


Minor comments:

- While Equation 16 is easy to follow, there is a big jump in Equation 17.
- In section 4.3, "Since h_K(K_j) does change for the queries". Perhaps "Since h_K(K_j) does NOT change for the queries"?


Refs:

Treviso, M., Góis, A., Fernandes, P., Fonseca, E., & Martins, A. F. (2021). Predicting Attention Sparsity in Transformers. arXiv preprint arXiv:2109.12188.


Update:

I thank the authors for their responses and efforts to improve the paper. The newly added experiments alongside the discussion addressed my concerns. I believe this paper provides a new angle to study the efficiency of bucket-based methods, proposing a method that overcomes issues found in previous approaches. Therefore, the paper presents a step forward for this field, and I recommend it to be accepted.

**Summary Of The Paper:**

This paper introduces a new method based on learnable hash functions to reduce the O(N^2) cost of self-attention in transformers to O(N^1.5). As previously known for bucket-based approaches, this cost is only achieved when buckets are balanced. The paper investigates the effectiveness of related approaches regarding bucket imbalance issues by showing statistics for several attention heads of a pre-trained transformer.  A precise metric is designed to quantify this notion of efficiency ("attention utility"), and later this metric is optimized by learning separate parameterized hash functions for queries and keys. To be able to optimize this metric, the authors use the unbiased approximation to the softmax function proposed by Performer (Choromanski et al., 2020) as a regularization term. Experiments on several NLP and CV tasks show that the proposed method achieves better results than previous fast-transformers while being faster than a standard transformer.


**Summary Of The Review:**

Overall, I think this paper is well-written, has a clear and practical motivation, and provides an elegant solution to the quadratic bottleneck issue in transformers. Most of my concerns are about the lack of correctness in some parts of the paper, such as choosing hyperparameters or evaluating LSH attention. However, the authors can easily address these concerns in the rebuttal period. Therefore, I am learning towards acceptance.

---

> ### Author Response · Authors · 2021-11-15
> **Response to Reviewer zEkG (1/2)**
>
> We greatly thank the reviewer's appreciation and comments on our work, especially bringing up the work by Treviso et al.
>
> > “The idea of having a learnable function prior to bucketing is not entirely novel (see a contemporary idea by Treviso et al., 2021), but the formulation is concise and clear, enabling generalizations to previous approaches.”
>
> We have added a discussion and comparison with SparseFinder (Treviso et al., 2021), which learns the routing projections of queries and keys with positive and negative samples. This optimization objective can be seen as a weakly-supervised learning problem. Notice that the bucketing strategies in SparseFinder still belong to the ANN type of approach. In contrast, our LHA explicitly optimizes the bucketing strategy (i.e., the learnable hash functions) towards maximizing the attention utility, as correctly stated by the reviewer as a “precise metric designed to quantify the sparse attention efficiency”.
>
> > “The core of the proposed method relies on optimizing the attention utility, which relies on the Performer's softmax kernel to regularize the training regime. The paper opts to use a KL between the softmax kernel and LHA to that end. It would be interesting to see how this KL term evolves during training.”
>
> We have added a plot of the dynamics of the KL terms during training and analyzed the results in Section F of the appendix.
>
> > “The choice of 10 heads from the 3rd layer in section 3.2 looks arbitrary. In fact, it is widely known that different heads at different layers focus on distinct tokens. Thus, what is the motivation for that choice? Moreover, from which pre-trained model were the results extracted? This is not clear in the paper.”
>
> We randomly picked the 3rd layer as an example to show the imbalance problem in LSH-based sparsification of Transformer attentions.  In fact, such imbalance issues commonly exist in Transformer attention heads. We already provided the statistics of the bucket sizes and query-key ratios in our paper (page 3). We also conducted an additional analysis on the bucket imbalance issues in Section G of the appendix, where we plot the histograms of bucket sizes and query-key ratios for all $16 \times 10 = 160$ attention heads in a pre-trained Transformer.
>
> The configuration of the pre-trained Transformer model is given in Section A.1 of the appendix. We have made this point more clearly in the revised version.
>
> > “Figure 1 shows the bucket size of queries and keys for each attention head using LSH attention. Since LSH attention was originally designed to use tied queries and keys (i.e., queries = keys), how can queries and keys have different bucket sizes? Also, it is unclear if queries and keys were tied for all experiments using LSH attention in the rest of the paper.”
>
> The reviewer is correct that Reformer (using LSH) does tie the queries and keys.  However, in our analysis/experiments in this paper, we run LSH without tying queries and keys and thus in settings different from Reformer (page 3).  We choose this way because we want to develop an efficient attention model that can be used as a plug-and-play replacement of dense attention layers without imposing the tie between queries and keys (page 3). We have made this more clear in the revised version.
>
> > “Which setting regarding the number of attention heads and layers was used for experiments on the GLUE benchmark and LRA? That is, how many layers and heads were set to use LHA?”
>
> The settings of GLUE and LRA are already given in Section A.4 and A.5 of the appendix, respectively.
>
> For the GLUE benchmark, all the attention heads are set to use LHA (i.e., variant (c), as mentioned on page 9). For the LRA benchmark, half of the attention heads in each layer are set to use LHA (i.e., variant (d), as mentioned on page 9).

---

> ### Author Response · Authors · 2021-11-15
> **Response to Reviewer zEkG (2/2)**
>
> > “In section 3.2, it is said that "queries should attend to enough keys to get a good approximation of the full-attention.". Is there a precise notion of what is "enough" in this context? That is, for recovering the true softmax distribution, queries should attend to ALL keys. In contrast, if we aim to recover an entmax distribution, queries can attend to only a subset of keys (see the sparse-consistency property in Treviso et al., 2021). So, what would change in LHA if we want to approximate entmax instead of softmax?”
>
> On the one hand, when ignoring the weights in the attention map, we use query-key ratios to measure whether queries attend to enough keys. As unbalanced query-key ratios would decrease the number of connections between queries and keys.
>
> On the other hand, in this paper, we mainly consider the case of preserving more weights in the attention map rather than more unweighted connections. So we can use the proposed attention utility metric, which describes how many *salient* keys are preserved for each query.
>
> The analysis in the paper about attention utility and learnable hash functions can be seamlessly applied to entmax. As for LHA with entmax, the only problem is that there is currently no unbiased kernelized approximation to entmax, so the training efficiency will increase from O(n^1.5) to O(2) (i.e., the same as SparseFinder), but the inference efficiency will still be O(n^1.5).
>
> > “While Equation 16 is easy to follow, there is a big jump in Equation 17.”
>
> We have added a complete derivation of our LHA objective in Section E of the appendix.

---

> > ### Comment · Reviewer_zEkG · 2021-11-20
> > **Response to Authors**
> >
> > > For the LRA benchmark, half of the attention heads in each layer are set to use LHA (i.e., variant (d), as mentioned on page 9).
> >
> > In Table 1, the variant (d) uses 0 NL layers. Also, variant (e) uses 0 NL heads. What is the meaning of having 0 NL layers/heads? Is it a full transformer?
> >
> >
> > > We have added a plot of the dynamics of the KL terms during training and analyzed the results in Section F of the appendix.
> >
> > The plots look great! The trade-off between the diversity and KL divergence shows right away the impact of having a learnable LSH. Great job! Just out of curiosity, why do the curves grow so abruptly exactly after step 1000?
> >
> >
> > > (about tying Qs and Ks in Reformer) We have made this more clear in the revised version.
> >
> > Where exactly? I believe that explicitly saying "In our analysis/experiments in this paper, we run LSH without tying queries and keys" somewhere in the paper would facilitate reproducibility and improve readability.
> >
> >
> > > So we can use the proposed attention utility metric, which describes how many salient keys are preserved for each query.
> >
> > This is not what the attention utility says. For a given cluster assignment, it describes the total amount of probability mass given by the connections of queries and keys inside the clusters, right?
> >
> > A small yet important note: if entmax (or sparsemax) attention is used instead of softmax, it is much easier to find an optimal assignment for Eq. 6. Namely, any assignment that keeps $Q_i$ and $K_j$ connected when $\mathrm{entmax}_{ij} > 0$ is going to maximize attention utility. Nonetheless, I believe the problem is still NP-hard.

---

> > > ### Author Response · Authors · 2021-11-20
> > > **Response to Reviewer zEkG**
> > >
> > > We appreciate the reviewer's response very much! Many thanks for your careful review again.
> > >
> > > > In Table 1, the variant (d) uses 0 NL layers. Also, variant (e) uses 0 NL heads. What is the meaning of having 0 NL layers/heads? Is it a full transformer?
> > >
> > > We thank the reviewer for pointing it out. Variant (d) uses NL attention in half of the attention heads in each layer, while variant (e) sets all the attention heads in the top 8 layers to NL attention. We have revised the manuscript to make the notations of NL heads and NL layers more consistent in the table.
> > >
> > > > The plots look great! The trade-off between the diversity and KL divergence shows right away the impact of having a learnable LSH. Great job! Just out of curiosity, why do the curves grow so abruptly exactly after step 1000?
> > >
> > > We don’t have any hyperparameters same as or close to 1000 in the experiments. One possible reason is that step 1000 just happens to be long enough for queries and keys to learn to construct more diverse attention patterns rather than nearly uniform attention.
> > >
> > > > Where exactly? I believe that explicitly saying "In our analysis/experiments in this paper, we run LSH without tying queries and keys" somewhere in the paper would facilitate reproducibility and improve readability.
> > >
> > > On page 3, we mentioned that “We use the same LSH technique as in Reformer, except that we do not impose extra constraints to queries and keys. This is because we would like to develop a plug-and-play replacement for dense attention layers without imposing extra constraints for queries and keys.”
> > >
> > > > This is not what the attention utility says. For a given cluster assignment, it describes the total amount of probability mass given by the connections of queries and keys inside the clusters, right?
> > >
> > > Yes, the reviewer’s understanding is correct. The attention utility precisely describes the aggregation of weights (saliency) of the keys in the same bucket for each query, which is only an approximation of "how many salient keys are preserved".

---

### Author Response · Authors · 2021-11-15
**Summary of Revisions**

We greatly appreciate reviewers for the effort spent on reviewing our work. Learning from the insightful and valuable suggestions, we summarize our major revisions to the paper as follows.

1. In Section 2, we addressed Reviewer zEkG’s comment with a discussion comparing LHA with SparseFinder [1], a contemporary work.


2. In Section E of the appendix, we addressed the concerns of Reviewer zEkG and Reviewer fhAP with a complete derivation of the LHA objective.


3. In Section F of the appendix, we addressed the Reviewer zEkG’s suggestion with analysis of the ​​training dynamics of the KL divergence (i.e., the optimization objective of learning-to-hash) and bucket-wise attention utility.


4. In Section G of the appendix, we addressed the Reviewer wDdo's suggestion with additional analysis on the training & inference efficiency of LHA. We found that LHA can achieve more significant speedup when facing longer sequences.


5. In Section H of the appendix, we addressed the Reviewer fhAP's suggestion with additional analysis on the bucket imbalance issues, i.e., plotting the histograms of bucket sizes and query-key ratios of different sparse patterns for all 16 × 10 = 160 attention heads in a pre-trained Transformer. Our results show that learning-to-hash attention (l2h) has more buckets close to the optimal bucket size and has more balanced query-key ratios for hash buckets, thus alleviating the bucket imbalance issues.


[1] Treviso, M., Góis, A., Fernandes, P., Fonseca, E., & Martins, A. F. (2021). Predicting Attention Sparsity in Transformers. arXiv preprint arXiv:2109.12188.

---

### Decision · Program_Chairs · 2022-01-20

**Decision:**

Accept (Poster)

**Comment:**

This paper adds to the literature of efficient sparse attention for long-range transformer architectures: a learned hash function is proposed by building successfully upon contributions from previous work. A similar idea appears in contemporary work, but with clear and complementary differences.

The reviewers are convinced of the importance of attention complexity, bucket imbalance issues, and agree that learning-to-hash is is a promising solution. The authors have clarified almost all outstanding concerns, in some cases adding valuable new results (e.g. timing experiments.)

I echo the reviewers' concern and stress to the authors to clarify the precise meaning of "plug-and-play", as it may be misinterpreted (e.g., no fine-tuning needed? or just no change to model but still fine-tuning is needed.)

Some of the cited papers are accepted at conferences, please update your bib file with the correct information for credit attribution.